# Profiling of G-Protein Coupled Receptors in Adipose Tissue and Differentiating Adipocytes Offers a Translational Resource for Obesity/Metabolic Research

**DOI:** 10.3390/cells12030377

**Published:** 2023-01-19

**Authors:** Saeed Al Mahri, Meshail Okla, Mamoon Rashid, Shuja Shafi Malik, Jahangir Iqbal, Maria Al Ibrahim, Ghida Dairi, Amer Mahmood, Manikandan Muthurangan, Ahmed Yaqinuddin, Sameer Mohammad

**Affiliations:** 1Experimental Medicine Department, King Abdullah International Medical Research Center (KAIMRC), King Saud Bin Abdulaziz University for Health Sciences (KSAU-HS), Ministry of National Guard Health Affairs (NGHA), Riyadh 11426, Saudi Arabia; 2College of Medicine, Alfaisal University, Riyadh 11533, Saudi Arabia; 3Department of Community Health Sciences, College of Applied Medical Sciences, King Saud University, Riyadh 11495, Saudi Arabia; 4Stem Cell Unit and Anatomy Department, College of Medicine, King Saud University, Riyadh 11495, Saudi Arabia; 5Biostatistics and Bioinformatics Department, King Abdullah International Medical Research Center (KAIMRC), King Saud Bin Abdulaziz University for Health Sciences (KSAU-HS), Ministry of National Guard Health Affairs (NGHA), Riyadh 11426, Saudi Arabia; 6King Abdullah International Medical Research Center (KAIMRC), King Saud Bin Abdulaziz University for Health Sciences (KSAU-HS), King Abdulaziz Medical City Hospital, Ministry of National Guard Health Affairs (NGHA), Al Hasa 31982, Saudi Arabia; 7Physiology Department, College of Medicine, King Saud University, Riyadh 11362, Saudi Arabia; 8Deanship of Scientific Research, Umm Al-Qura University, Makkah 21961, Saudi Arabia

**Keywords:** adipose tissue, adipogenesis, G-protein-coupled receptors, thermogenesis, obesity, metabolic syndrome

## Abstract

G protein-coupled receptors (GPCRs) are expressed essentially on all cells, facilitating cellular responses to external stimuli, and are involved in nearly every biological process. Several members of this family play significant roles in the regulation of adipogenesis and adipose metabolism. However, the expression and functional significance of a vast number of GPCRs in adipose tissue are unknown. We used a high-throughput RT-PCR panel to determine the expression of the entire repertoire of non-sensory GPCRs in mouse white, and brown adipose tissue and assess changes in their expression during adipogenic differentiation of murine adipocyte cell line, 3T3-L1. In addition, the expression of GPCRs in subcutaneous adipose tissues from lean, obese, and diabetic human subjects and in adipocytes isolated from regular chow and high-fat fed mice were evaluated by re-analyzing RNA-sequencing data. We detected a total of 292 and 271 GPCRs in mouse white and brown adipose tissue, respectively. There is a significant overlap in the expression of GPCRs between the two adipose tissue depots, but several GPCRs are specifically expressed in one of the two tissue types. Adipogenic differentiation of 3T3-L1 cells had a profound impact on the expression of several GPCRs. RNA sequencing of subcutaneous adipose from healthy human subjects detected 255 GPCRs and obesity significantly changed the expression of several GPCRs in adipose tissue. High-fat diet had a significant impact on adipocyte GPCR expression that was similar to human obesity. Finally, we report several highly expressed GPCRs with no known role in adipose biology whose expression was significantly altered during adipogenic differentiation, and/or in the diseased human subjects. These GPCRs could play an important role in adipose metabolism and serve as a valuable translational resource for obesity and metabolic research.

## 1. Introduction

The G-protein coupled receptors (GPCRs) are the largest family of membrane proteins and are expressed widely in the body. They regulate several important biological processes, and dysregulation of GPCR signaling has been implicated in the pathogenesis of many diseases [1,2,3] There are ~800 members of the GPCR family and more than 400 are sensory receptors (olfactory, vision, and taste receptors). The remaining ~356 are non-sensory receptors and are activated by physical ligands [4]. The unique blend of variety and specificity within the GPCR family and the fact that they are readily targetable by exogenous drugs make GPCRs attractive therapeutic targets [5,6]. Roughly 30–40% of the currently marketed drugs target GPCRs. However, the overall number of targeted GPCRs is about 30% and a vast number of GPCRs are yet to be exploited for therapeutic purposes. In adipose tissue, GPCRs regulate critical processes such as adipogenesis, lipolysis, thermogenesis, glucose metabolism, and secretion of adipokines [7,8,9,10]. Of all the GPCRs, β-adrenergic receptors (βARs) have been comprehensively studied for their role in adipose metabolism [11,12,13,14]. All three members of the βAR family (β1, β2, and β3) are expressed in adipose with β3 AR being the predominant one. These βARs are stimulated by norepinephrine and signaling occurs via the activation of G-protein subunit Gαs leading to the accumulation of second messenger cAMP that phosphorylates and activates its downstream targets including cAMP-dependent protein kinase A (PKA). Activation of PKA leads to increased lipolysis via hormone-sensitive lipase (HSL), adipose triglyceride lipase (ATGL), and perilipin [11]. βARs are also involved in thermogenesis in brown adipose tissue [15]. In addition to the βARs, several other GPCRs have been shown to play important functional roles in white and brown fat including alpha-adrenergic receptors [16], free fatty acid receptors [7,17,18], adenosine receptors [19], adhesion receptors [20,21], hydroxycarboxylic acid receptors [22,23], and many others [9]. These still represent a significantly small fraction of the total number of GPCRs, and the functional significance of a vast majority of GPCRs is not known in adipose metabolism. This is to a certain extent due to the inadequate data regarding their expression in adipose tissue and differentiating adipocytes. Therefore, this study was initiated to comprehensively profile the entire repertoire of non-visual GPCRs, using high-throughput RT-PCR in mouse brown and white adipose tissue and during adipogenic differentiation of 3T3-L1 cells. In addition, the expression of GPCRs in subcutaneous adipose tissues from healthy lean, healthy obese and unhealthy obese human subjects as well as in adipocytes from mice fed a normal or high-fat diet was assessed by re-analyzing RNA-sequencing data [24,25]. Our comprehensive analysis revealed several GPCRs with no known role in adipose biology that were highly expressed in adipose tissue and demonstrated significant changes in expression during adipogenesis and/or in diseased human subjects. Understanding the role of these recently discovered GPCRs in the metabolism of adipose tissue could lead to the discovery of previously unrecognized disease-drug relationships and speed up the process of developing new drugs for obesity and metabolic disorders.

## 2. Materials and Methods

### 2.1. Reagents and Chemicals

Mouse 3T3-L1 cell lines were purchased from American Type Culture Collection (ATCC), (Manassas, VA, USA). The cell culture reagents including Advanced Dulbecco’s modified Eagle’s medium (DMEM), GlutaMAX, Penicillin-Streptomycin (10,000 U/mL), and fetal bovine serum (FBS) were ordered from Gibco Life Technologies (Thermo Fisher Scientific, Waltham, MA, USA). The Oil Red O solution, 3-isobutyl-1-methylxanthine (IBMX), dexamethasone, Rosiglitazone, and insulin were obtained from Sigma-Aldrich (St. Louis, MO, USA). PureLink RNA purification kit by Ambion, cDNA synthesis kit, TaqMan Array Mouse GPCR Panel, advanced TaqMan PCR Master Mix, and SYBR Green PCR Master Mix were purchased from Applied Biosystem (Thermo Fisher Scientific, Waltham, MA, USA). DNA oligonucleotides were obtained from Macrogen. Antibodies were obtained from Thermo Fisher (Thermo Fisher Scientific, Waltham, MA, USA) and Cell Signaling Technology (Danvers, MA, USA).

### 2.2. Culture and Adipogenic Differentiation of 3T3-L1 Adipocytes

Cells were maintained in the complete advanced DMEM media until confluence. Two to three days post-confluence, the differentiation was induced using an adipogenic cocktail which included 0.25 µM dexamethasone, 0.5 mM IBMX, 3 µM Rosiglitazone, and 1 μM insulin in advanced DMEM with 10% FBS. Four days post differentiation the culture medium was changed with advanced DMEM + 10% FBS+ 1µM Insulin for 24 h (post-differentiation). Cells were maintained in advanced DMEM with 10% FBS and used on Day 7 of differentiation.

### 2.3. Isolation of Mouse Adipose Tissue Depots

All protocols and procedures were approved by the Institutional Research Ethics Committee (REC) at King Saud University in Riyadh, Saudi Arabia (approval no. KSU-SE-18-39). C57BL/6 male mice from Taconic Biosciences, USA, were housed in a room temperature set at 22–24 °C and 45% humidity under 12 h light/dark cycles daily. At 10 weeks of age, 5 mice were sacrificed and both inguinal subcutaneous white adipose tissue (WAT) and interscapular brown adipose tissue (BAT) were collected, snap-frozen in liquid nitrogen, and stored at −80 °C until analysis.

### 2.4. RNA Isolation and RT-PCR by Taqman Arrays

Total RNA was extracted from mouse adipose tissue depots using Trizol reagent and isolated using the chloroform-isopropanol method. To remove genomic DNA contamination RNA was purified using DNase (5 PRIME). RNA was isolated from cultured 3T3L1 cells using an RNeasy kit with DNase treatment (Qiagen, Valencia, CA, USA). Reverse transcription was performed using the High-Capacity cDNA Reverse Transcription Kit (Applied Biosystems, Waltham, MA, USA). Gene expression array was performed on cDNA samples from 3T3-L1 cells and mouse adipose tissue depots, and GPCR expression was determined with TaqMan Array Mouse GPCR Panel (Applied Biosystems, Waltham, MA, USA) according to the recommended protocol. The arrays were run on an ABI Prism 7900HT system (Applied Biosystems), and analyzed with the Sequence Detection System software (Applied Biosystems, Waltham, MA, USA). These arrays are efficient, easy-to-use microfluidic cards for quantitative gene expression analysis of GPCR genes. The expression of 352 non-visual GPCRs was determined with these arrays. GPCR expression was quantified relative to the 15 housekeeping genes (Actb, B2m, Gapdh, Gusb, Hmbs, Hprt1, Ipo8, Pgk1, Polr2a, Ppia, Rplp2, Tbp, Tfrc, Ubc, Ywhaz) in the sample. Based on the relative expression, GPCR mRNA levels were sub-divided as “Expressed”, “Trace” and absent/non-quantifiable as described by Patricio Atanes et al. [26]. Genes in the expressed category were present > 0.1% of the mRNA expression of 15 housekeeping genes, while GPCRs with a relative expression between 0.01% to 0.1% were present at trace levels. Those GPCRs that were either not detected or present at <0.01% of endogenous controls were considered “Absent”. This roughly translates to Ct values of <30, 30–33, and >33 for “Expressed”, “Trace” and “Absent” GPCRs, respectively.

### 2.5. Western Blotting

Cell pellets were lysed in RIPA buffer (50 mM Tris; pH 7.5, 150 mM NaCl, 0.5% sodium deoxycholate, 0.1% SDS, 2 mM EDTA, and 1% Triton X-100) containing protease inhibitor (Halt protease inhibitor cocktail, Thermo Fisher Scientific, Waltham, MA, USA). The lysates were incubated on ice for 15 min followed by the centrifugation for 15 min at 15,000× *g* at 4 °C. The clear lysate was transferred to a new 1.5 mL Eppendorf tube and protein quantification was done using Qubit Protein Assay Kit (Thermo fisher Scientific, Waltham, MA, USA). Separation of proteins by SDS-PAGE, immunoblotting was done with the indicated primary antibodies, and secondary HRP-conjugated antibodies using enhanced chemiluminescence detection reagents.

### 2.6. RNA Sequencing Re-Analysis

RNA sequencing datasets GSE152991 and GSE142187 were analyzed by the GEO RNA-seq Experiments Interactive Navigator (GREIN) platform [27]. This dataset contained RNA sequencing data of subcutaneous adipose tissue from 45 human subjects. This includes metabolically healthy lean (MHL) (*n* = 11) had BMI 18.5–24.9 kg/m^2^, plasma TG concentration <150 mg/dL, fasting plasma glucose concentration <100 mg/dL, 2-h OGTT plasma glucose concentration <140 mg/dL, HbA1c ≤5.6%, and IHTG content <4%. Metabolically healthy obese (MHO) (*n* = 15) had BMI 30–49.9 kg/m^2^, plasma TG concentration <150 mg/dL, fasting plasma glucose concentration <100 mg/dL, 2-h OGTT plasma glucose concentration <140 mg/dL, HbA1c ≤5.6%, and IHTG content <4%. Metabolically unhealthy obese (MUO) (*n* = 20) had BMI 30–49.9 kg/m^2^, prediabetes (fasting plasma glucose concentration ≥100 mg/dL, 2-h OGTT plasma glucose concentration ≥140 mg/dL, and/or HbA1c ≥5.7%), and IHTG content ≥5% [24]. Dataset GSE142187 contained RNA expression data from adipocytes isolated from white adipose tissue of mice fed either a regular chow or high-fat diet for 20 weeks [25]. CPM-normalized mRNA expression was obtained from GREIN) platform and differential gene expression was estimated. Raw *p* values were adjusted for multiple testing using the Benjamini–Hochberg procedure and only adjusted *p* (Padj) values higher than 0.05 were considered significant. The expression of 356 non-visual GPCRs was determined in these samples [28].

### 2.7. Statistical Analyses

Data are presented as the mean  ±  standard error of mean (SEM) values. Two-tailed Student’s *t*-test was used to determine significant differences. * *p* < 0.05; ** *p* < 0.01; *** *p* < 0.001.

## 3. Results

### 3.1. GPCR Expression Profile in Mouse White and Brown Adipose Tissue

Taqman qPCR arrays were able to detect of 292 GPCRs in mouse white adipose tissue. Of these, 267 were expressed above trace levels, 25 at trace levels, and 60 were not detected by the arrays (Figure 1A). On the other hand, brown adipose tissue expressed 271 GPCRs of which 196 were detected above trace levels, 75 were detected at trace levels, and 81 were absent or undetected (Figure 1B). The majority of GPCRs were expressed in both WAT and BAT but a significant number were differentially expressed in only one of the two adipose tissue types (Figure 1C). Lysophosphatidic acid receptor 6 (Lpar6), also known as P2ry5, was the highest expressed receptor in both WAT and BAT followed by frizzled Class Receptor 4 (Fzd4). Adhesion receptor 5 (Adgrf5) and orphan receptor Gprc5b were also highly expressed in both adipose tissue types (Figure 2A,B). GPCRs that are expressed in both adipose tissue depots but enriched in WAT include chemokine receptors (Cxcr5 and Ccr6), mas-related G-protein coupled receptor member D (Mrgprd), relaxin family peptide receptor 3 (Rxfp3) and 5-hydroxytryptamine (serotonin) receptor 1F (Htr1f) (Figure 2C). Those enriched in BAT include leucine-rich repeat-containing G-protein coupled receptor 6 (Lgr6), galanin receptor 1 (Galr1), secretin receptor (SCTR), free fatty acid receptor 4 (Ffar4), sphingosine-1-phosphate receptor 5 (S1pr5) and P2Y purinoreceptor 1 (P2ry1) (Figure 2D). A number of GPCRs are exclusively expressed in one of two adipose depots and absent/undetected in the other. The complete list of GPCRs that are expressed in only one of the two adipose tissue depots is shown in Table 1.

### 3.2. Comparative Analysis of GPCR mRNA Expression in Undifferentiated and Differentiated 3T3-L1 Adipocytes

The 3T3-L1 cells were differentiated into adipocytes using the mentioned cocktail. Oil red O, which stains neutral lipids and triglycerides, was used to validate differentiated adipocytes. Oil red O stained multiple droplets in differentiated adipocytes compared to pre-adipocytes that did not show any lipid droplets (Appendix A). Additional validation of the differentiation process was done by evaluating the mRNA and protein expression of key adipogenic markers (Appendix A).

In undifferentiated 3T3-L1 pre-adipocytes, TaqMan Arrays detected 250 GPCRs of which 214 were detected above trace levels, 36 were expressed at trace levels and 102 were absent/undetected. On day 2 of differentiation 234 GPCRs were detected, of which 123 were expressed at trace levels, 111 were expressed above trace levels, and 118 were undetected/absent. Fully differentiated 3T3-L1 adipocytes (day 7) expressed a total of 231 GPCRs of which 132 were expressed above trace levels, 99 at trace levels, and 121 were undetected or absent (Figure 3A). The Venn diagrams in Figure 3B show these three groups with stratification of GPCR mRNA expression and the extent of commonality in 3T3-L1 cells at different stages of differentiation. The highly expressed GPCRs in 3T3-L1 cells at different stages of differentiation are shown in Figure 3C. Fzd1, Secreted Frizzled Related Protein 2 (Sfrp2), Calcitonin Receptor-like Receptor (Calcrl), Lpar1, and orphan receptor (Gpr153) are among the highly expressed GPCRs in undifferentiated 3T3-L1 cells (day 0). Gprc5b, Lgr4, SMO (smoothened), Fzd1, and Fzd4 are among the highly expressed GPCRs on Day 2. The list of GPCRs highly expressed in differentiated adipocytes (Day 7) includes Gprc5b, F2r (Coagulation Factor II Thrombin Receptor), Gpr137, Fzd4, and Lgr4 (Figure 3C). Adipogenic differentiation of 3T3-L1 cells profoundly impacts the expression GPCRs as shown in the volcano plots (Figure 4A). The expression of a large number of GPCRs is downregulated, whereas fewer GPCRs show an increased expression during the differentiation process. GPCRs that undergo significant upregulation during adipogenesis include Ffar4, Gprc5c (orphan GPCR), Gipr (Gastric inhibitory peptide), Hcar2, Gpr35, and Adrb2. Highly downregulated GPCRs include Secreted frizzled-related proteins (Sfrp1, Sfrp2, and Sfrp3), orphan GPCRs (Gpr176 and Gpr149), Neuropeptide Y Receptor Y1 (Npy1R), and Gpr153. Heat maps depict the highly upregulated GPCRs (Figure 4B) and highly downregulated GPCRs (Figure 4C) during the differentiation process. GPCRs that are upregulated during the adipogenesis process and their functional significance in adipose are shown in the Table 2.

GPCR expression in human subcutaneous adipose tissue and the impact of obesity RNA sequencing detected a total of 257 GPCRs in subcutaneous adipose of healthy human subjects, of which 172 were expressed above trace levels, 85 were expressed at trace levels whereas 99 GPCRs were absent or not detected (Figure 5A). In terms of relative expression, the most abundant GPCRs include Frizzled Class Receptor 4 (FDZ4), Adhesion G Protein-Coupled Receptor (ADGRF5, ADGRL2, ADGRL4, ADGRA2), Prostaglandin E Receptor 3 (PTGER3), Calcitonin receptor-like (CALCRL), Neuropeptide Y receptor Y1 (NPY1R) and Leucine-Rich Repeat Containing G Protein-Coupled Receptor 4 (LGR4) (Figure 5B). Obesity has a substantial effect on the expression of GPCRs, and several members of this family show significant changes in their expression in both healthy and unhealthy obese human subjects (Figure 5C). Heat maps show the top 10 GPCRs that are downregulated (Figure 6A) or upregulated (Figure 6B) in healthy obese and unhealthy obese subjects. The list of highly upregulated GPCRs includes Chemokine receptors (CCR5, C3AR1, CCR1), GPR183, Formyl peptide receptor (FPR3), GPR132, and adhesion GPCR (ADGRG4). Top downregulated GPCRs in obese subjects include Calcitonin receptor-like (CALCRL), Prokineticin receptor 1 (PROKR1), GPR146, LGR4, Adhesion G Protein-Coupled Receptors (ADGRL2 and ADGRB3), GIPR, Angiotensin II Receptor Type 1 (AGTR1), arginine vasopressin receptor 1A (AVPR1A) and ADRβ3.

### 3.3. GPCR Expression in Adipocytes Isolated from Normal and High-Fat Fed Mice

GSE142187 RNA sequencing dataset was re-analyzed to assess the expression of GPCRs in adipocytes from normal lean and diet-induced obese mice. The dataset comprised RNA expression data of adipocytes extracted from white adipose tissue of mice fed a regular or high-fat diet for 20 weeks [26]. The GPCR expression profile resembled that of mature 3T3-L1 adipocytes, and the majority of the highly expressed GPCRs were found in both mouse adipocytes and cultured 3T3-L1 adipocytes (Figure 7A). High-Fat diet had a substantial effect on the expression of GPCRs, and several members of this family show significant changes in their expression (Figure 7B). Most of the GPCRs exhibit expression changes that are comparable to those observed in human obesity (upregulated or downregulated in both HFD mice and obese human individuals), whereas only a few GPCRs show differences between the two groups. The detailed comparison is shown in Appendix A.

### 3.4. GPCRs of Interest

Our comprehensive analysis revealed several GPCRs that had remarkable expression profile, which includes: [1] high expression in human adipose tissue; [2] significant changes in the diseased state; and/or [3] significant upregulation during the adipogenic differentiation of 3T3 L1 cells. GPCRs whose expression is significantly altered in adipose tissue of obese human subjects are listed in Table 3 (downregulated) and Table 4 (upregulated). Many GPCRs in this list have previously been studied for their role in adipose biology. These include alpha- and beta-adrenergic receptors [11,12,13,14,15,16], adenosine receptors [19], free fatty acid receptors [7,17,18], adhesion receptors [20,21], hydroxycarboxylic acid receptors [22,23], gastric inhibitory peptide [27,28,29], chemokine receptors [30,31] and many others [32,33,34,35]. Interestingly, many GPCRs with notable expression profiles in adipose tissue and adipocytes have no known role in adipose metabolism. Here is the description of these important GPCRs that warrant further research to investigate their adipose-specific functional significance and therapeutic potential.

#### 3.4.1. GPRC5 Family of Receptors

The GPCR5 group of receptors are part of the class C GPCR family and consist of four members GPRC5A, GPRC5B, GPRC5C, and GPRC5D. These are orphan GPCRs, and their expression is induced by retinoic acid. GPR5A is significantly upregulated during the adipogenesis process (Figure 8A) and is expressed at low levels in mouse adipose tissue (Figure 8B). GPCR5A is highly expressed in human adipose and is downregulated in obese subjects (Figure 8C). GPRC5B and GPRC5C are significantly upregulated during adipogenic differentiation of 3T3 L1 adipocytes (Figure 8D,G) and are highly expressed in mice (Figure 8E,H) and human adipose tissue (Figure 7F,I), suggesting their possible role in adipose function. GPRC5D is expressed at trace levels in human adipose but absent in mouse adipose tissues and adipocytes (data not shown).

We also checked the protein expression of Gprc5b and Gprc5c during the adipogenic differentiation of 3T3-l1 cells and consistent with the RT-PCR data; protein expression of both Gprc5b and Gprc5c showed a marked increase in mature adipocytes (Day 7) as compared to pre-adipocytes (Day 0) (Figure 9).

#### 3.4.2. G Protein-Coupled Receptor 146 (GPR146)

GPR146 was significantly upregulated during the adipogenic differentiation of 3T3 L1 cells (Figure 10A). It was highly expressed in mouse (Figure 10B) and human adipose tissue (Figure 10C) and significantly reduced in both healthy and unhealthy obese subjects (Figure 10C). This makes GPR146 a good candidate to be studied for its possible adipose-specific role.

#### 3.4.3. G Protein-Coupled Receptor 137 (GPR137)

In this study we observed that GPR137 is highly upregulated during the adipogenesis process (Figure 11A). It is also highly expressed in mice (Figure 11B) and human adipose tissue (Figure 11C). These observations indicate a possible role of GPR137 in adipose tissue metabolism.

#### 3.4.4. Arginine Vasopressin Receptor 1A (AVPR1A)

AVPRR1A was detected at very low levels in 3T3L1 cells (Figure 12A) and in mouse adipose tissue depots (Figure 12B) in our study. However, AVPR1A was highly expressed in human adipose tissue and was significantly downregulated in obese subjects (Figure 12C). This strongly suggests a role for AVPR1A in human adipose tissue function and possible therapeutic value.

## 4. Discussion

GPCRs play a significant role in regulating adipose biology by regulating processes like adipogenesis, thermogenesis, insulin sensitivity, and glucose metabolism. However, only a few members have been comprehensively characterized and studied for their role in adipose metabolism. One of the reasons for this is the lack of information about their expression in adipose tissue and differentiating adipocytes. GPCRs are expressed at relatively lower levels, and therefore, previous techniques were not sensitive enough to detect a majority of the members of this family. Previous reports have demonstrated that RNA-seq and TaqMan GPCR arrays are much better suited than transcriptomic cDNA microarrays for assessing GPCR expression [28]. Due to their low abundance, Affymetrix arrays detect fewer GPCRs as compared to the Taqman arrays and RNA sequencing. We generated the expression profile of the entire set of non-visual GPCRs by using a high-throughput RT-PCR Array GPCR panel in mouse adipose tissue and differentiating adipocytes and re-analyzing RNA sequencing data of adipose tissue from healthy and obese human subjects. Many of the GPCRs with striking expression profiles have been comprehensively characterized for their adipose-specific functions and have attracted considerable attention in the drug discovery field [7,8,9,10,11,12,13,14,15,16,17,18,19,20,21,22,23]. Chemokines and chemokine receptors have long been associated with obesity-linked inflammation, and insulin resistance and modulation of the chemokine/Chemokine receptor axis is considered a novel approach for the treatment of obesity and associated metabolic disorders [39,40]. Similarly, previous studies have reported increased expression of NPYR1 expression in adipose tissue of obese human subjects and high-fat diet-fed mice [41,42]. Moreover, selective antagonism of peripheral NPYR1 was shown to prevent insulin resistance and other metabolic abnormalities in high-fat diet-fed mice [41,42,43,44]. ADRβ3 and GIPR are thought to provide great therapeutic opportunities in obesity and type 2 diabetes [11,12,29]. However, ADRβ3 and GIPR expression is markedly down-regulated in adipose tissue of obese human subjects in this study. This is consistent with other studies that have reported downregulation of these receptors in the adipose tissue of high fat-fed mice as well as in human obese subjects [45,46]. Therefore, the approach to using agonists of these receptors to mitigate obesity and related disorders is unlikely to work. This makes it even more imperative to discover newer GPCR-based targets that could offer anti-obesity therapeutic opportunities.

Our extensive analysis identified many GPCRs with striking expression profiles that merit a comprehensive functional characterization in adipose tissue metabolism. These include GPRC5 Family of receptors (GPRC5A, GPRC5B, and GPRC5C), orphan receptors GPR146 and GPR137 and Arginine vasopressin receptor 1A (AVPR1A). GPRC5B has previously been shown to activate obesity-induced inflammatory signaling in adipocytes and Gprc5b-/- mice are protected from diet-induced obesity and insulin resistance [38]. In humans, GPRC5B is expressed ubiquitously and a genome-wide association study has identified GPRC5B as a genetic locus for obesity predisposition, probably due to copy-number variance [47]. The functional significance of GPRC5A and GPRC5C in adipose tissue metabolism is not known, and would be an exciting research subject. GPRC5A has been extensively investigated in cancer cells for its role as a tumor suppressor. In cancer cells, GPRC5A acts as a negative regulator of PI3K/AKT and cAMP/PKA pathways, and it remains to be seen whether GPRC5A regulates these pathways in adipose tissue [48]. GPR146 is an orphan receptor expressed widely in humans. Several studies have indicated that GPR146 is a putative receptor for C-peptide [49]. Previous reports have demonstrated that GPR146 plays a vital role in the regulation of plasma cholesterol levels [49,50,51]. A non-coding variant rs1997243 regulates blood cholesterol levels through the upregulation of GPR146 expression and GPR146-/- mice are protected from hypercholesterolemia and atherosclerosis [52]. Therefore, antagonism of GPR146 has been suggested as a potential strategy to tackle atherosclerotic cardiovascular disease. The role of GPR146 in adipose tissue metabolism has not been studied. GPR137 is an orphan receptor widely expressed in human tissues and has been previously studied for its role in cancer cell growth and proliferation [53,54]. GPR137 is highly expressed in several cancer types and is directly involved in cancer cell proliferation and metastasis. No previous study has investigated the role of GPR137 in adipose metabolism. AVPR1A is the receptor for arginine vasopressin (AVP) peptide and abundantly expressed in the central nervous system, heart, and liver. A wide variety of behaviors, including stress management, territorial aggressiveness, social bonding, and recognition, have been shown to be modulated by the AVP/AVPR1A signaling pathway [55,56]. The AVPR1A receptors signal by activation of phospholipase C and increased intracellular calcium, which, in turn, stimulates vasoconstriction [57,58]. AVPR1A expression is increased in failing hearts, and overexpression of AVPR1A in a mouse model has been shown to impair cardiac function [58]. Other GPCRs with remarkable expression profiles include orphan GPCR (GPR35) [32], Prokineticin Receptor 1 (PROKR1) [59], LGR4 [60,61], and Adhesion G Protein-Coupled Receptor F5 (ADGRF5) [20]. Previous studies have suggested that these GPCRs may play a critical role in controlling adipocyte biology and systemic energy homeostasis but more work is needed for a detailed understanding of their functional significance in adipose tissue metabolism. Overall, this study adds a new layer of complexity to the already intricate repertoire of genes that play crucial roles in adipose metabolism. Therefore, improving our understanding of the biological functions of such receptors in adipose tissue has clinical relevance and may prove essential in the drug discovery process.

## 5. Conclusions

In conclusion, our comprehensive analysis identified several highly expressed GPCRs with no known role in adipose biology whose expression was significantly altered during adipogenic differentiation and/or in the diseased human subjects. Understanding the functional importance of these GPCRs in adipose tissue metabolism might aid in the uncovering of hidden disease–drug connections and speed up drug development for obesity and metabolic diseases.

The limitations of this study include using whole adipose tissue for profiling GPCR expression. Adipose tissue is not a homogenous tissue and contains adipocytes and a stromal vascular fraction (SVF) of cells including pre-adipocytes, fibroblasts, vascular endothelial cells, and a variety of immune cells such as adipose tissue macrophages. The cellular composition of adipose tissue may change in obesity, which may have a direct impact on the expression of genes including GPCRs. We analyzed a dataset (GSE #142287) comprising of adipocytes isolated from white adipose tissue of mice fed either a regular chow or a high-fat diet for 20 weeks. High-Fat diet had a significant impact on the expression of GPCRs which was largely comparable to those observed in human obesity. However, only single-cell RNA sequencing will be helpful in delineating the cell-specific expression of GPCRs in healthy and diseased states. In addition, the study relies solely on mRNA levels that may or may not reflect the changes in the protein levels. As well as the protein expression, the cellular localization of GPCRs is key to their functioning. Hence, it is important to determine the protein levels and subcellular localization of individual GPCR by immunoblotting and immunohistochemistry.

## Figures and Tables

**Figure 1 cells-12-00377-f001:**
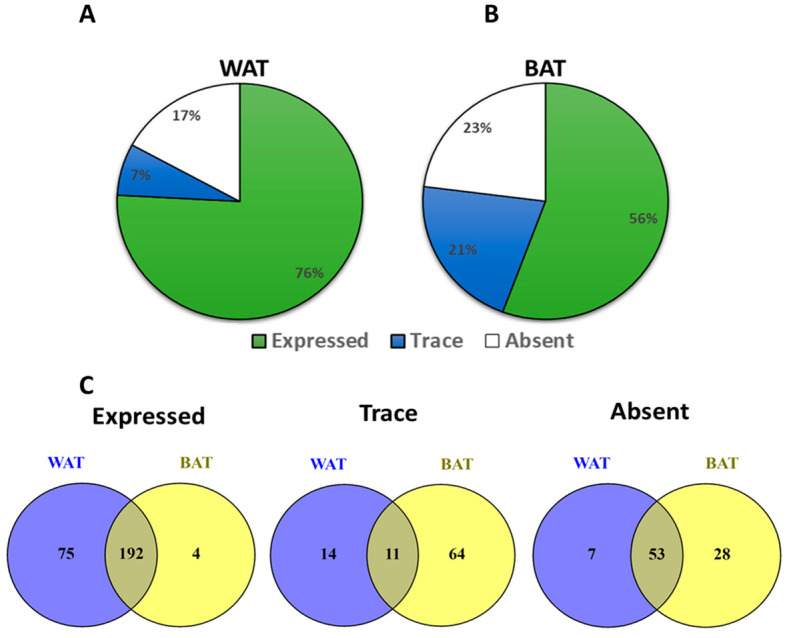
GPCRs expression in mouse White Adipose Tissue (WAT) and Brown Adipose Tissue (BAT). (**A**,**B**) Pi chart showing the proportion of expressed, trace and absent/undetected GPCRs in mouse WAT and BAT. (**C**) Venn diagrams showing the relationship (distinct or overlapping) between the expressed, traced and absent/undetected GPCRs identified in WAT and BAT mouse tissues. Each Data point represents an average mRNA expression of two independent experiments. The data are normalized to the average of 15 reference genes in the same sample as described in the method section.

**Figure 2 cells-12-00377-f002:**
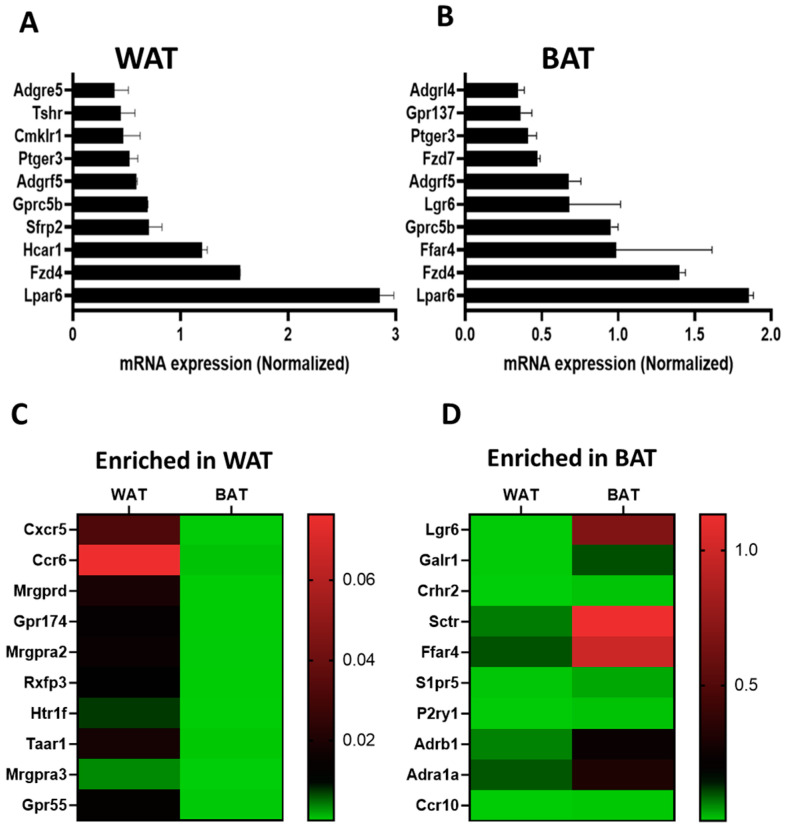
Comparison of highly expressed GPCRs between WAT and BAT. (**A**,**B**) mRNA expression of the top 10 expressed GPCRs in WAT and BAT. (**C**,**D**) Heat maps showing normalized gene expression profiles for ten most enriched GPCRs in WAT and BAT. Data were generated using pooled RNA of WAT and BAT samples obtained from five mice. Each Data point represents an average mRNA expression of two independent experiments. The data are normalized to the average of 15 reference genes in the same sample as described in the method section.

**Figure 3 cells-12-00377-f003:**
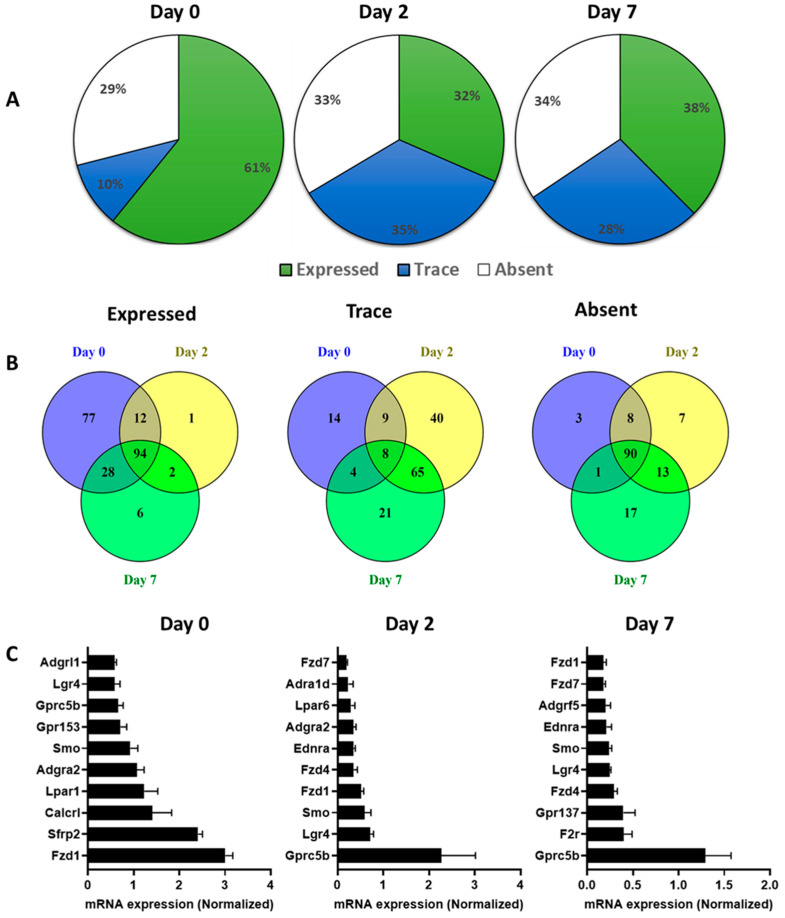
The GPCR mRNA expression during the adipogenic differentiation of 3T3 L1 cells. (**A**) The fraction of the expressed, trace or absent GPCRs in the pre-adipocytes (Day 0), early adipocytes (Day 2) and mature adipocytes (Day 7). (**B**) Venn diagrams showing the exclusive or shared genes for expressed, trace or absent mouse GPCR mRNAs from at day 0 (Blue), at day 2 (Yellow) and at day 7 (Green). (**C**) mRNA expression of the top 10 expressed GPCRs at Day 0, Day 2 and DAY 7 of differentiating 3T3 L1 cells. Each Data point represents an average of three independent experiments. The data are normalized to the average of 15 reference genes in the same sample as described in the method section.

**Figure 4 cells-12-00377-f004:**
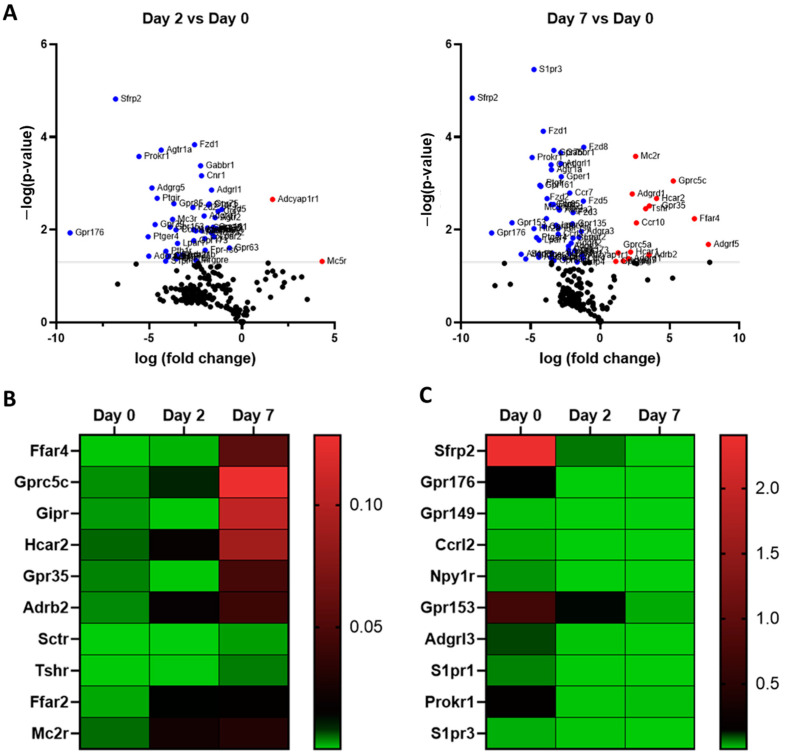
Differential expression of GPCRs in 3T3 L1 cells before and after differentiation (**A**) Volcano plot showing differentially expressed GPCRs genes, with -log (*p*-values) plotted against log (fold-changes) in Day2 and Day7 as compared to Day 0. GPCRs shown in color show a significant difference in expression with −Log (*p*-values) more than 1.3 which corresponds to *p*-values < 0.05. The GPCRs shown in red are upregulated whereas those in blue are downregulated (**B**,**C**) Heat maps showing normalized expression values for the 10 most downregulated and upregulated GPCRs during the differentiation of 3T3 L1 adipocytes. Each Data point represents an average of three independent experiments. The data are normalized to the average of 15 reference genes in the same sample as described in the method section.

**Figure 5 cells-12-00377-f005:**
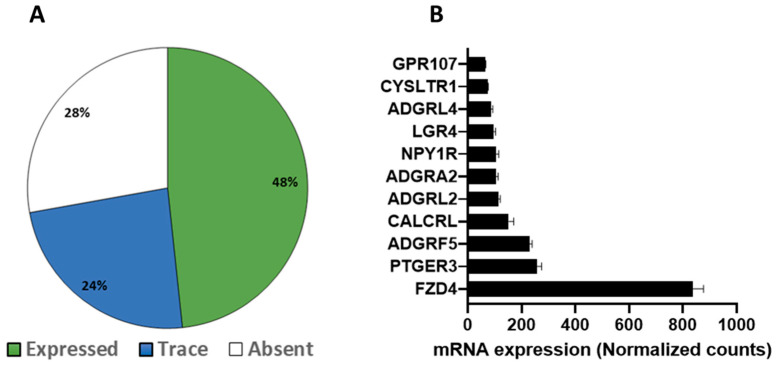
The GPCR mRNA repertoire in the subcutaneous adipose tissue of healthy human subjects from the RNA sequencing data set GSE 152,991 (**A**) The proportion of the expressed, trace and undetected GPCRs and (**B**) the most abundant GPCRs in the subcutaneous adipose of healthy human subjects. Each data point represent normalized mRNA counts from 11 healthy lean human subjects.

**Figure 6 cells-12-00377-f006:**
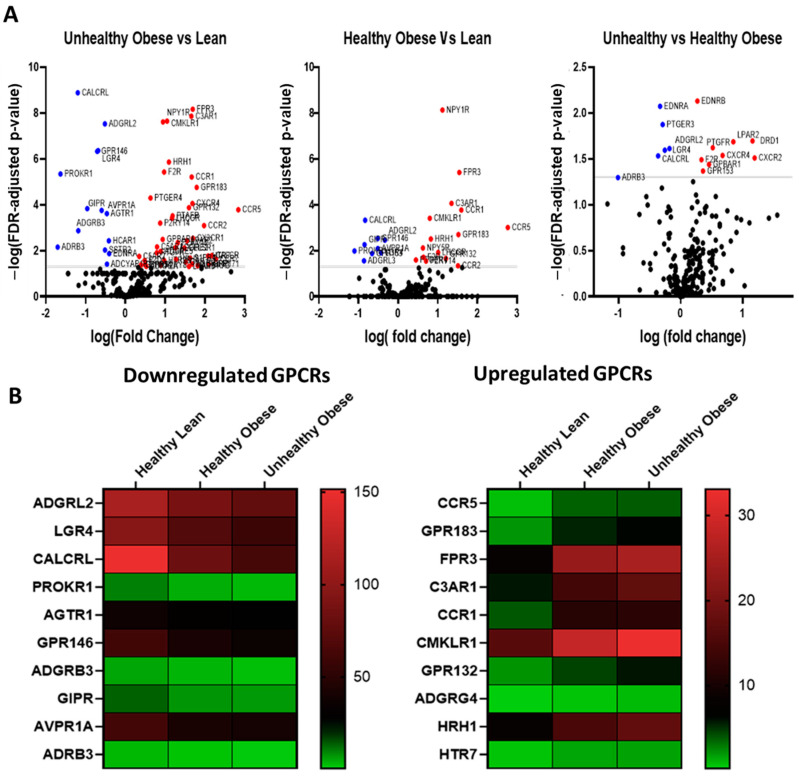
The GPCR mRNA repertoire in the subcutaneous adipose tissue of healthy lean, healthy obese and unhealthy obese human subjects. (**A**) Volcano plots showing differentially expressed GPCRs (FDR adjusted *p*-value < 0.05) in color, with FDR plotted against fold-change in subcutaneous adipose of unhealthy obese subjects versus lean, healthy obese versus lean, and unhealthy obese against healthy obese subjects. The GPCRs shown in red are upregulated whereas those in blue are downregulated. (**B**) Heat maps showing individual expression values for the 10 most downregulated and upregulated GPCRs in subcutaneous adipose tissue of human subjects without or with obesity. Each data point represent normalized mRNA counts from healthy lean (N = 11), healthy obese (N = 14), and unhealthy obese (N = 20) as described in the method section.

**Figure 7 cells-12-00377-f007:**
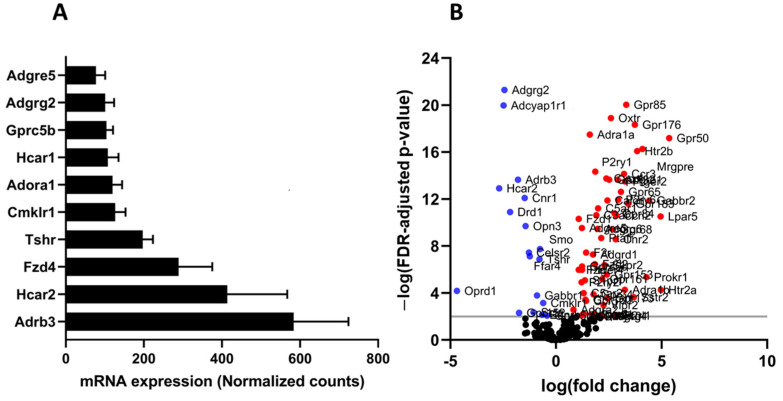
The GPCR mRNA repertoire in mouse adipocytes. (**A**) the most abundant GPCRs in the adipocytes isolated from healthy mouse adipose tissue and (**B**) Volcano plots showing differentially expressed GPCRs (FDR adjusted *p*-value < 0.01) in color, with FDR plotted against fold-change in adipocytes isolated from white adipose tissue of mice fed a high fat diet versus regular chow fed mice. The GPCRs shown in red are upregulated whereas those in blue are downregulated. Each data point represent normalized mRNA counts from 6 animals in each group.

**Figure 8 cells-12-00377-f008:**
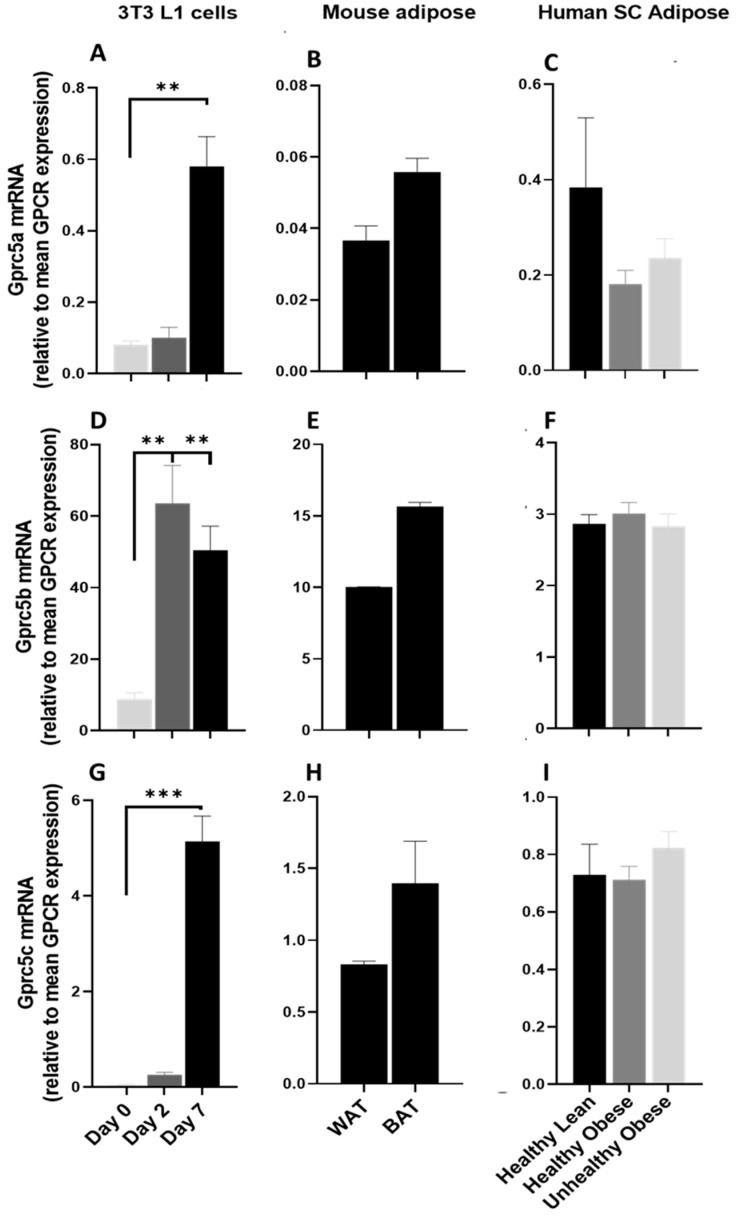
The mRNA expression of GPRC5 family in 3T3 L1 cells, mouse adipose depots and human adipose tissue. Changes in the mRNA expression of Gprc5A, Gprc5b and Gprc5c (**A**,**D**,**G**) during the adipogenic differentiation of 3T3 L1 cells. Each bar represents an average of three independent experiments (**B**,**E**,**H**) WAT and BAT from pooled tissue samples from 5 mice. Each graph represents an average of two independent experiments (**C**,**F**,**I**) human subcutaneous adipose tissue of lean and obese human subjects. Each data point represent normalized mRNA counts from healthy lean (N = 11), healthy obese (N = 14) and unhealthy obese (N = 20). The mRNA expression is shown relative to average mRNA expression of all detected GPCRs. Date is expressed as average ± SEM and statistical analysis by unpaired *t*-test. Double Asterisk, *p* < 0.01 and triple Asterisk, *p* < 0.001.

**Figure 9 cells-12-00377-f009:**
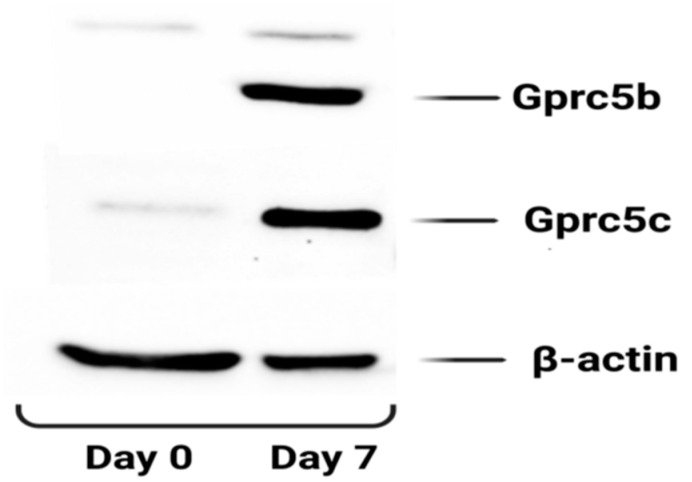
The protein expression of Gprc5b and Gprc5c in 3T3 L1 cells before and after differentiation. The blots were probed with the indicated antibodies. β-actin was used as a loading control. The blot is a representative image of two independent experiments.

**Figure 10 cells-12-00377-f010:**
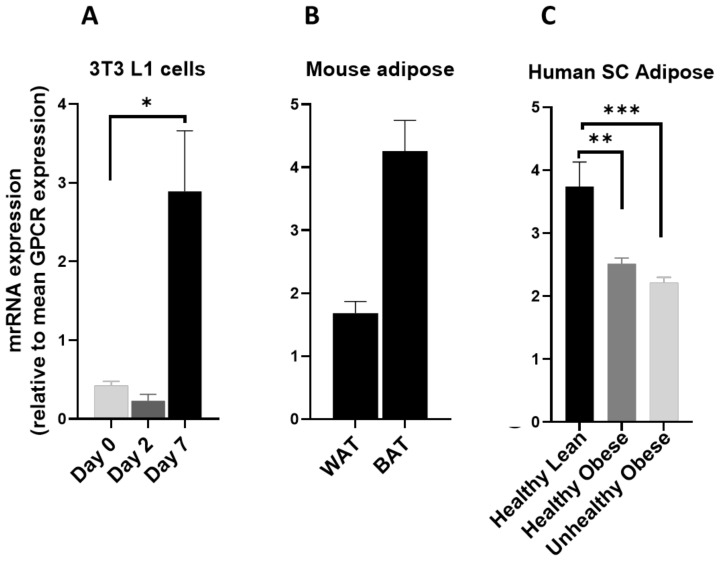
GPR146 mRNA in 3T3 L1 cells, mouse adipose depots and human adipose tissue. GPR146 mRNA expression (**A**) during the adipogenic differentiation of 3T3 L1 cells. Each bar represents an average of three independent experiments (**B**) WAT and BAT from pooled tissue samples from 5 mice. Each graph represents an average of two independent experiments (**C**) human subcutaneous adipose tissue of lean and obese human subjects. Each data point represent normalized mRNA counts from healthy lean (N = 11), healthy obese (N = 14) and unhealthy obese (N = 20). The mRNA expression is shown relative to average mRNA expression of all detected GPCRs. Date is expressed as average ± SEM and statistical analysis by unpaired *t*-test. Asterisk, *p* < 0.05, double Asterisk, *p* < 0.01 and triple Asterisk, *p* < 0.001.

**Figure 11 cells-12-00377-f011:**
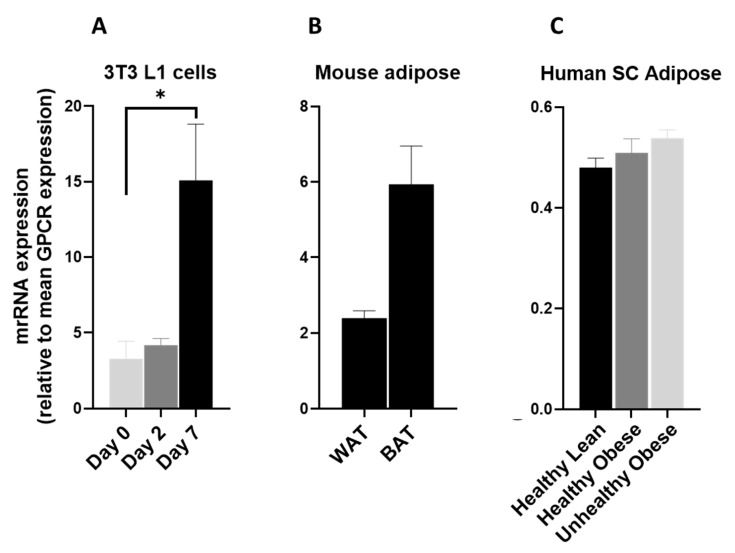
GPR137 mRNA in 3T3 L1 cells, mouse adipose depots and human adipose tissue. GPR137 mRNA expression (**A**) during the adipogenic differentiation of 3T3 L1 cells. Each bar represents an average of three independent experiments (**B**) WAT and BAT from pooled tissue samples from 5 mice. Each graph represents an average of two independent experiments (**C**) human subcutaneous adipose tissue of lean and obese human subjects. Each data point represent normalized mRNA counts from healthy lean (N = 11), healthy obese (N = 14) and unhealthy obese (N = 20). The mRNA expression is shown relative to average mRNA expression of all detected GPCRs. Date is expressed as average ± SEM and statistical analysis by unpaired *t*-test. Asterisk, *p* < 0.05.

**Figure 12 cells-12-00377-f012:**
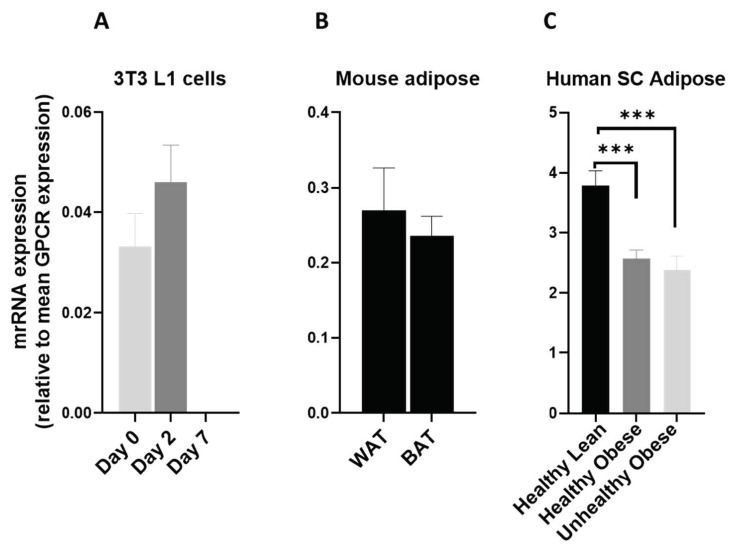
AVPR1A mRNA in 3T3 L1 cells, mouse adipose depots and human adipose tissue. AVPR1A mRNA expression (**A**) during the adipogenic differentiation of 3T3 L1 cells. Each bar represents an average of three independent experiments (**B**) WAT and BAT from pooled tissue samples from 5 mice. Each graph represents an average of two independent experiments (**C**) human subcutaneous adipose tissue of lean and obese human subjects. Each data point represent normalized mRNA counts from healthy lean (N = 11), healthy obese (N = 14) and unhealthy obese (N = 20). The mRNA expression is shown relative to average mRNA expression of all detected GPCRs. Date is expressed as average ± SEM and statistical analysis by unpaired *t*-test. Triple Asterisk, *p* < 0.001.

**Table 1 cells-12-00377-t001:** List of GPCRs that are only detected in one of the two mouse adipose tissue depots (WAT or BAT).

Only in WAT	Only in BAT
Mrgprb3	Lpar1
Mrgpra1	Tacr2
Ffar3	Adgrf3
Mrgpra3	Nmbr
Gpr12	Adgrb3
Ltb4r2	Mchr1
Gpr119	Drd2
Npy2r	
Gpr6	
Tacr1	
Npy5r	
Npbwr1	
Oxgr1	
Rgr	
Grm4	
Gpr83	
Hrh4	
Hcrtr2	
Ptgdr	
Gpr139	
Glp2r	
Npsr1	
Cckar	
Tacr3	
Glp1r	
Galr3	
Grm6	
Nmur1	

**Table 2 cells-12-00377-t002:** List of GPCRs that are significantly upregulated during adipogenic differentiation of 3T3-L1 cells and their functional significance in adipose tissue biology.

Name	Log Fold Change	Role in Adipose	References
Adora1	7.865386	Induces lipolysis and increases energy expenditure	[19]
Adgrf5	7.774302	promotes insulin signaling and insulin-mediated glucose uptake	[20]
Ffar4	6.765778	Enhances insulin sensitivity, increases glucose uptake and thermogenesis.	[7]
Gprc5c	5.247853	Unknown	
Gipr	5.197849	Enhances insulin sensitivity, increases glucose uptake and lipoprotein lipase activity.	[29,30,31]
Hcar2	4.04649	Unknown	
Gpr35	3.526624	Regulates lipid metabolism, thermogenic, and anti-inflammatory gene expression in adipose tissue	[32]
Adrb2	3.493813	Regulates lipolysis and thermogenesis	[11,12,13,14]
Sctr	3.402992	Induces lipolysis	[33]
Tshr	3.266825	Regulates BAT adipogenesis	[34]
Ffar2	2.625755	Regulated adipogenesis	[7,35]
Ccr10	2.592673	Unknown	
Mc2r	2.541011	Unknown	
Mc5r	2.435111	Promotes lipolysis and impairs re-esterification in adipocytes	[36]
Adgrd1	2.292871	Unknown	
Hcar1	2.172346	Inhibits lipolysis and stimulates glucose uptake	[22,23]
Adgrg1	2.029279	Regulated adipogenesis	[20]
Adgrg7	1.64874	Unknown	
Gprc5a	1.279647	Unknown	
Gpr146	1.114973	Unknown	
Kiss1r	1.020995	Promotes adipocyte differentiation and fat accumulation	[37]
Gprc5b	0.974769	Activates obesity-associated inflammatory signaling in adipocytes	[38]
Gpr17	0.948663	Unknown	
F2r	0.738862	Unknown	
Gpr137	0.717915	Unknown	

**Table 3 cells-12-00377-t003:** List of GPCRs that are downregulated in the subcutaneous adipose tissue of obese human subjects and their expression profile in mouse adipose tissue and 3T3 L1 adipocytes.

GPCR Name	Log Fold Change (Human Adipose)		Log Fold Hange 3T3-L1	Expression in Mouse Adipose		Previously Studied in Adipose
	Healthy Obese vs. Lean	Unhealthy Obese vs. Lean	Day 7 vs. Day 0	WAT	BAT	
GPR50	−0.680	−2.517	Not detected	No	No	Yes
GPR179	−0.657	−2.153	Not detected	No	No	No
ADRB3	−0.759	−1.694	2.36	Yes	Yes	Yes
PROKR1	−1.054	−1.666	−4.89	Yes	Yes	yes
SSTR1	−0.387	−1.322	−1.81	Yes	Yes	Yes
ADGRB3	−0.635	−1.237	Not detected	No	Yes	No
CALCRL	−0.835	−1.206	−3.56	Yes	Yes	Yes
GIPR	−0.838	−0.955	5.19	Yes	Yes	Yes
GPR146	−0.520	−0.725	1.11	Yes	Yes	No
CHRM3	−0.441	−0.709	−2.7	Yes	Yes	No
LGR4	−0.428	−0.695	−1.26	Yes	Yes	Yes
GPR63	−0.590	−0.653	−0.05	Yes	Yes	No
AVPR1A	−0.521	−0.608	ND	Yes	Yes	No
ADGRL2	−0.341	−0.528	−2.28	Yes	Yes	No
SSTR2	−0.314	−0.519	0.67	Yes	Yes	No
AGTR1	−0.309	−0.489	−3.50	Yes	Yes	NO
ADCYAP1R1	−0.441	−0.473	−1.30	Yes	Yes	Yes
HCAR1	−0.341	−0.426	2.17	Yes	Yes	Yes
EDNRA	−0.080	−0.418	−0.57	Yes	Yes	Yes

**Table 4 cells-12-00377-t004:** List of GPCRs that are upregulated in the subcutaneous adipose tissue of obese human subjects and their expression profile in mouse adipose tissue and 3T3 L1 adipocytes.

GPCR Name	Log Fold Change Human Adipose		Log Fold Change 3T3-L1	Expression in Mouse Adipose		Previously Studied in Adipose
	Healthy Obese vs. Lean	Unhealthy Obese vs. Lean	Day 7 vs. Day 0	WAT	BAT	
C3AR1	1.306	3.748	−0.67	Yes	Yes	Yes
CCR5	2.391	2.494	ND	Yes	Yes	Yes
CCR7	1.51	2.377	−2.20	Yes	Yes	Yes
ADGRG3	0.876	2.258	−2.72	Yes	Yes	Yes
FPR1	0.93	2.117	−1.83	Yes	Yes	Yes
CXCR2	0.956	1.989	−2.16	Yes	Yes	Yes
CCR2	1.485	1.959	−2.10	Yes	Yes	Yes
GPR183	1.463	1.725	−1.52	Yes	Yes	Yes
CXCR4	1.053	1.702	ND	Yes	Yes	Yes
CCR1	1.596	1.67	−1.66	Yes	Yes	Yes
FPR3	1.524	1.666	ND	No	No	No
HTR7	1.611	1.655	ND	Yes	Yes	No
S1PR4	0.864	1.633	−2.53	Yes	Yes	No
ADGRE1	0.919	1.623	ND	Yes	Yes	No
CX3CR1	1.285	1.622	−1.97	Yes	Yes	Yes
GPR132	1.141	1.534	−2.08	Yes	Yes	Yes
ADORA3	1.057	1.44	ND	Trace	Trace	No
LPAR2	0.611	1.432	−1.53	Trace	Trace	Yes
CELSR1	1.058	1.379	ND	Yes	Yes	No
HRH2	0.806	1.354	−2.10	Yes	Yes	Yes
LPAR5	0.841	1.334	−1.91	Yes	Yes	No
P2RY13	0.754	1.314	−2.04	Yes	Yes	No
FPR2	0.713	1.295	−2.27	Yes	Trace	Yes
GPR85	1.038	1.253	−3.55	Yes	Yes	No
ADGRE2	0.74	1.227	ND	No	No	No
DRD1	−0.073	1.211	−1.53	Yes	trace	Yes
GPR162	0.923	1.2	−1.40	Yes	Yes	No
LHCGR	1.004	1.151	ND	Yes	Yes	No
PTAFR	0.77	1.148	ND	Yes	Yes	Yes
HRH1	0.833	1.084	−1.63	Yes	Yes	No
CMKLR1	0.809	1.047	−2.51	Yes	Yes	Yes
OPRK1	1.717	1.014	ND	No	No	Yes

## Data Availability

The datasets used and/or analyzed during the current study are available from the corresponding author on reasonable request.

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
