# Peer review of "Profiling of G-Protein Coupled Receptors in Adipose Tissue and Differentiating Adipocytes Offers a Translational Resource for Obesity/Metabolic Research"

_cells, 2023, doi:10.3390/cells12030377_

Round 1

Reviewer 1 Report

This paper is fine for Cells.

Author Response

Thank you

Reviewer 2 Report

Mahri et al. used a high-throughput RT-PCR panel to determine the expression of the entire repertoire of non-sensory GPCRs in mouse white, and brown adipose tissue and assess changes in their expression during adipogenic differentiation of murine adipocyte cell line, 3T3-L1. The expression of GPCRs in subcutaneous adipose tissues from lean, obese, and diabetic human subjects was also evaluated by re-analyzing RNA-sequencing data. They detected a total of 292 and 271 GPCRs in mouse white and brown adipose tissue, respectively. Results showed that there is a significant overlap in the expression of GPCRs between the two adipose tissue depots but several GPCRs are specifically expressed in one of the two tissue types. They reported several highly expressed GPCRs with no known role in adipose biology whose expression was significantly altered during adipogenic differentiation and/or in the diseased human subjects. Their conclusion demonstrated that GPCRs could play an important role in adipose metabolism and serve as a valuable translational resource for obesity and metabolic research. 

I offer some following minor corrections and suggestions:

1.     Words in the Table 1 are overlapped, author should correct them.

2.     In the Figure S1A, scale bar should be added.

3.     A small font size is more difficult to read. Author should adjust the size of the text in the figure to make sure it is easy to read and navigate (Figure 1, Figure S1, Figure 2, Figure 3)

Author Response

Thanks for reviewing our paper. here is the point-by-point response to your queries.

  1. Words in the Table 1 are overlapped, the author should correct them.

A- Corrected

  1. In the Figure S1A, scale bar should be added.

A- Done, the scale bar has been added to Figure S1

  1. A small font size is more difficult to read. Author should adjust the size of the text in the figure to make sure it is easy to read and navigate (Figure 1, Figure S1, Figure 2, Figure 3)

A- The figures have been rearranged and the font size increased to make sure it is easy to read and navigate

Reviewer 3 Report

This manuscript by Mahri er al. aims at characterize the GPCRome in adipose tissue and adipocytes. GPCRs are targets for ~35% of approved drugs but only 32 ~15% of the ~800 human GPCRs are currently such targets. Therefore, new opportunities likely exist for GPCRs

as drug targets. Given the crucial role of the adipose tissue and adipocytes for proper metabolic function profiling the GPCRome here could reveal new therapeutic targets. Unfortunately, the paper is very descriptive and at current stage only a ressource with limited use. This is because the GPCRome has been performed either on a) Whole-tissue samples b) Immortalized cell line. From my perspective, this paper could be an interesting and very relevant ressource for the adipose field, however, it requires a major revision with new data. 

Major points:

  1. Its a big limitation that the GPCRome is done on whole adipose tissue. The adipose tissue is composed of several  different cell types and the cellular composition changes between adipose depots and physiological conditions. With the current data/analysis the expression of a given GPCR is an average expression across multiple cell types in the adipose tissue. 
  2. The use of 3T3-L1 cells suggest that the authors want to profile the GPCRome in mature adipocytes. However, 3T3-L1 cells are not a suitable or physiological relevant model for this. I suggest that the authors are floating/isolating mature adipocytes from adipose tissue and do a GPCRome analysis to answer point 1. and 2. This will also help focusing the current manuscript and significanly increase the novelty. Alternatively, the authors could do computational deconvulation to map the expression individual GPCRs across all cell types. 
  3. The authors could do a more in deepth analysis of current data (and potential new data) to increase the impact furher. For example, a. describe the precentage of GPCRs coupling to e.g. Gs, Gq etc. b. Is there any SNPs associated with adipose tissue-related diseases? c. Ligand-Receptor interaction - which cell type(s) produces the ligand for the GPCRs.
  4. Confirmation on protein level of selected differentially expressed GPCRs (Western blotting, RNAscope, IHC etc). This confirmation could be done on floated mature adipocytes to answer. Evidence for protein expression can also be obtained by assessment of signaling pathways and functional activity.

Minor:

  1. Table 2 needs references for indicated function in adipocytes. 
  2. Statistical analysis of Fig4B, Fig5B, Fig6B might be missing. 

Author Response

Thanks for reviewing the paper. Attached, please find our response to your queries. 

Round 2

Reviewer 2 Report

Authors have improved the manuscript significantly in regards to the original version. From my point of view, the new version of the manuscript that is including the answers to my questions and inquiries has been well performed. For me, this new version is including all indicated aspects for considering to be integrated.

Author Response

Thanks 

Reviewer 3 Report

I'm very pleased with the response to points 3) and 4) and the minor points. 

However, I don't think the authors are addressing points 1) and 2), which is the biggest limitation of the manuscript. Peers have already described the GPCRome composition of bulk adipose tissue in a variety of conditions (e.g. DOI: 10.1016/bs.mcb.2015.10.004DOI: 10.1038/s41366-020-0570-2DOI: 10.1016/j.pharmthera.2014.09.007DOI: 10.1016/j.cell.2008.08.040,DOI: 10.1016/j.cell.2021.04.037). Identifying cell type-specific expression profiles would increase the novelty of the manuscript tremendously. The authors can either do this by bioinformatic deconvolution, however, given the low expression of GPCRs in general this might not be the best method. Instead, I suggest that the authors are doing a GPCR profiling of floated/isolated mature adipocytes from adipose tissue (eventually +/- a physiological condition like HFD, cold, etc.). This will significantly increase the novelty as this has not been done before. Further, given that the authors then only are focusing on one cell type it will satisfy my criticism for points 1) and 2). 

Having said this, I appreciate that the authors are mentioning the limitations in the discussion. From my point of view, it must be an editorial decision whether it is enough to write this limitation in the discussion or whether it should be addressed experimentally. 

Author Response

Thank you for reviewing our paper. We included an analysis of RNA sequencing dataset (GSE142187). The study used adipocytes extracted from white adipose tissue (by fractionation) of mice fed either a conventional chow or high-fat diet. The expression profile of GPCRs in adipocytes was comparable to that of mature 3T3-L1 adipocytes employed in our investigation. Furthermore, a high-fat diet had a considerable effect on GPCR expression. Most GPCRs show expression changes comparable to those seen in human obesity (upregulated or downregulated in both HFD mice and obese humans), with just a few GPCRs showing variations between the two groups. Figure 7 and Table S1 illustrate the results. Our future research will concentrate on determining the cell-specific expression of GPCRs in human adipose tissue in both healthy and pathological conditions.